# Total Protein Content, Amino Acid Composition and Eating-Quality Evaluation of Foxtail Millet (*Setaria italica* (L.) P. *Beauv*)

**DOI:** 10.3390/foods12010031

**Published:** 2022-12-22

**Authors:** Siyu Hou, Yihan Men, Min Wei, Yijuan Zhang, Hongying Li, Zhaoxia Sun, Yuanhuai Han

**Affiliations:** 1College of Agriculture, Institute of Agricultural Bioengineering, Shanxi Agricultural University, Jinzhong 030801, China; 2Shanxi Key Laboratory of Minor Crops Germplasm Innovation and Molecular Breeding, Taiyuan 030031, China

**Keywords:** foxtail millet, protein, amino acid composition, eating quality

## Abstract

Foxtail millet has attracted substantial attention in recent years because of its excellent properties as a cereal crop with high nutritional value. Although the cultivation area of foxtail millet keeps growing, the fundamental research into the nutritional and eating qualities of foxtail millet germplasm collections is limited. In this study, we performed a survey of protein content, amino acid composition and eating quality among a germplasm collection of foxtail millet accessions grown in different environments. Our results revealed 21 accessions with stable protein content under different environments. The correlation analysis further revealed that the protein content of the grains was affected by environmental and genotypic interactions. The further amino acid composition analyses suggested that higher protein content accessions have a better essential amino acid index, providing more nutritional value for human beings and animal feedstock. Moreover, the flavor-related amino acid content and other eating-quality trait analyses were also performed. The subordinative analysis suggested that B331 could be the best accession with high protein content and superior eating quality. Taken together, this study provides essential nutritional and eating-quality data on our germplasm collection of foxtail millets, and provides a core genetic resource from which to breed elite foxtail millet varieties in the future.

## 1. Introduction

Foxtail millet (*Setaria italica* (L.) Beauv.), the most-grown millet in Asia, has a long cultivation history of more than two millennia in northern China, according to archaeological discoveries [1]. In recent years, foxtail millet has attracted substantial attention worldwide because of its many excellent properties such as self-pollination, being gluten-free, nutrient-rich, and stress tolerant [2]. Moreover, foxtail millet has been developed into a C4 model plant for fundamental research recently [3,4]. India and China are the top two countries for foxtail millet production with estimated production of 2.3 × 10^6^ and 0.4 × 10^6^ tons in 2020, respectively (FAO statistics, http://fao.org/faostat, accessed on 17 February 2022). The efforts of researchers and breeders to boost the production of foxtail millets highlight the benefits of leveraging this ancient grain to fight against global climate and population challenge [5].

The nutritional qualities, including protein content and composition, are important quality aspects of cereal crops. Amino acids are the building blocks of proteins. However, essential amino acids (EAAs) cannot be made in the human body and must come from food, because humans and animals lack the enzymatic machinery for the de novo synthesis of those amino acids. Only plants can synthesize all amino acids, including EAAs, which provide the only source of nitrogen in the human body [6]. As a kind of high dietary fiber whole-grain food, foxtail millet grains are also rich in protein, starch, vitamins (especially folates and provitamin A carotenoids) and minerals [7,8]. The protein content in the grains of foxtail millet range from 11.13 to 18.75% [9], which is relatively higher than rice (7%–9%), wheat (11%–14%), and maize (8%–11%). In mature foxtail millet grains, proteins are mainly in the form of prolamins (around 60%), a group of plant storage proteins with poor solubility in water, which is similar to that of other cereal grains [10]. Foxtail millet grains also have significant levels of EAAs with eight EAAs accounting for 44.70% of the total amino acid content [9]. Hence, foxtail millet can be used as an excellent protein source candidate.

Protein content and amino acid composition are the main quality attributes of foxtail millet, and these factors also provide important reference values for food processing, commodity ratings and variety breeding. Therefore, it is of great significance to develop a holistic grain quality evaluation framework based on protein content and amino acid composition. To date, only a handful studies have reported on the protein content/composition determination method, agronomic factors, and genes related to protein content in the grains of foxtail millet. A reference method for protein determination based on nitrogen analysis, also known as the Kjeldahl method, was first reported by Lynch and Barbano in 1999 [11]. This Kjeldahl method was also successfully applied to determine millet grain protein content and foxtail millet has the highest protein content (12.94 ± 0.87%) compared to finger millet and pearl millet [12]. To understand the complex matrix of starch and protein in grains, a near-infrared reflectance spectroscopy method was successfully adopted to calculate the relative content of amylose (0~25.6%), total starch (72.9~81.4%) and crude protein (8.9~16%) in foxtail millet [13]. Planting density is another agronomic factor determining crop yield and quality. Indeed, a field trial of foxtail millet showed that there were significantly higher grain yields (2227 Kg·ha^−1^) and protein content (increased 10.8%) in plants grown at a 20 cm × 10 cm plant spacing compared to those grown at other plant spacings [14]. To understand the constitution of seed storage proteins, a genome-wide prediction method was used and 47 seed storage proteins (SSPs) comprising 14 albumins, 12 prolamins, 18 globulins and 3 glutelins were identified in foxtail millet by the comparative analysis of 225 SSPs of rice, barley, sorghum and maize [15]. A recent investigation of the prolamin characteristics in their raw and post-cooked forms from different varieties suggested the effect of prolamins on foxtail millet palatability and also highlighted the importance of understanding the prolamin–starch interactions during the cooking process [16]. Interestingly, these differential proteins involved in amino acid metabolism were significantly upregulated and the protein contents were also improved under drought stress conditions in foxtail millet through comparative proteomic analysis [17]. In this study, we first investigated the total protein content (TPC) of more than 200 foxtail millet accessions in three field experiment environments using the Kjeldahl method. Ten accessions with stable protein content in all three environments were further subjected to amino acid composition and content analysis. According to the amino acid score (AAS) and EAA index (EAAI), the protein qualities of these accessions were determined, and the flavor-related amino acids in the grains of the different accessions of foxtail millet were explored. The correlation analyses of the protein content, starch content, grain hardness, saturated water absorptivity, gel consistency and grain color were performed, and the results give a hint to the further understanding of the complex relationship between those eating-quality traits in foxtail millet. Overall, this study provides a reference for protein quality, amino acid composition, and eating quality based on the survey of a foxtail millet germplasm collection. Furthermore, several accessions could be used as core genetic resources in future breeding projects aiming to increase eating quality.

## 2. Materials and Methods

### 2.1. Plant Materials and Growth Environment

In total, a germplasm collection of 320 foxtail millet accessions was provided by Shanxi Agricultural University and the Chinese Academy of Agricultural Sciences (CGRIS) as a kind donation (Appendix A). The experimental sites were located in Datong city (113°12′51″ E, 40°22′58″ N), Jinzhong city (112°35′21″ E, 37°25′16″ N) and Jincheng city (113°7′8″ E, 35°32′6″ N) in Shanxi Province, China, referred to as the E1, E2, and E3 planting environments, respectively. Sowing without fertilization was performed from 1 May to 10 May 2018 at the three experimental sites. The soil conditions including pH, soil organic matter, alkali-hydrolyzable nitrogen, available phosphorus, available kalium, Mn, Fe, Cu, Zn, Ca and Mg at the three experiment sites were detected by a reference standard assay method (Appendix A) [18]. The total field size was approximately 0.2 ha per experimental site with plots set in a randomized complete-block design. Each plot was 4 m × 1.5 m in dimension and comprised 5 rows of plants, 20 for each row. The row and plant spacing were set at 0.3 m and 0.2 m, respectively. The same plot setting and management were applied to all three experimental fields.

### 2.2. Total Protein Content (TPC) Determination

The mature grains were collected from 3 individual plants for each variety. Three independent biological repeats from the different plots were analyzed. The grains were ground into a powder using a grinding mill (Tissuelyser-24, Shanghai Jinxin Industrial Development Co., Ltd., Shanghai, China). The TPC of these samples was detected through the micro-Kjeldahl method with an automatic discrete chemistry analyzer [12] (SmartChem140, AMS-Alliance Co., Paris, France).

### 2.3. Amino Acid Composition and Evaluation of Nutritional Parameters

Two milligrams of defatted powder samples were hydrolyzed with 0.5 mL of 6 M HCl in a sealed ampoule containing 8 mL of phenol (to protect tyrosine from degradation) and 0.25 mL of norleucine (Cat. N8513, Sigma, as an internal standard) for 24 h at 110 °C under vacuum [19]. The acid hydrolysate was dried completely using a Speedvac concentrator (Senco, Shanghai Senco Technology Co., Shanghai, China), and the dried residue was redissolved in 0.5 mL citrate buffer. The sample was filtered through a 0.45 mm nylon filter before being analyzed with an automated amino acid analyzer (Hitachi 835-50, Hitachi Limited, Tokyo, Japan). Sulfur-containing amino acids, cystine and methionine were determined after prehydrolysis oxidation with performic acid. The contents of the recovered amino acids were presented as g/100 g protein and were compared with the Food and Agriculture Organization, the World Health Organization and the United Nations University (FAO/WHO/UNU) (2007) reference patterns (Appendix A). The essential amino acid (*EAA*) score was calculated by the FAO/WHO method as shown below [20]:(1)$$AAS\%=\frac{mg\;AA\;in\;1g\;of\;the\;protein\;tested}{mg\;AA\;in\;1g\;reference\;protein\;tested}$$
(2)$$EAAI\left(\%\right)=n\log EAA$$
where log *EAA* = [1/n] × [log (100 a1/a1R) + … + log (100 an/anR)]; a1 is the weight (mg) of amino acid in 1 g of tested protein; a1R is the weight (mg) of amino acid in 1 g of reference protein; and n is the number of amino acids considered for the calculation (the pair methionine-cysteine count as 1). *AAS* and *EAAI* represent amino acids and *EAA*s in Equations (1) and (2), respectively [19]. The reference protein used was the amino acid pattern defined by the WHO and FAO (FAO/WHO, 2013). The *AAS* of protein was considered the lowest *AAS* value among the *EAA*s.

### 2.4. Starch Content Analysis

The method for measuring starch was adapted from the modified Association of Official Agricultural Chemists (AOAC) method [21]. Briefly, 80% ethyl alcohol was added to a tube with 0.1 g of grain powder; the solution was stirred for 15 min to thicken after adding 2 mL of 1.7 mol/L NaOH. Then, 8 mL of 600 mM sodium acetate buffer (pH 5.0) and 0.1 mL of thermostable α-amylase (280 U) were added and mixed by vigorous vortex for 3~4 s; the samples were incubated at 50 °C for 30 min after adding 0.1 mL of amyloglucosidase (AMG, 330 U). The samples were centrifuged at 13,000× *g* rpm for 5 min. For each sample, 1 mL of supernatant was taken from the tube and diluted by 11-fold in 10 mL sodium acetate buffer for further analysis. The solution of 0.1 mL α-amylase and AMG, instead of buffer only, was used as a blank absorbance. For each measurement, 3 mL of GOPOD reagent (glucose oxidase/peroxidase) was added to 0.1 mL aliquots of diluted samples in a new tube and then was incubated at 50 °C for 20 min. The absorbance value was measured at the wavelength of 520 nm. The content of starch was calculated by the following formula:$$Starch\%=\Delta A\times F\times \frac{\mathrm{EV}}{0.1}\times D\times \frac{1}{1000}\times \frac{100}{W}\times \frac{162}{180}$$
where Δ*A*: absorbance value; *F*: factor to convert absorbance values to μg glucose; *D*: dilution ratio (11-fold); *W*: sample weight (g).

### 2.5. Gel Consistency

The gel consistencies of the grains of the 10 foxtail millet accessions were determined with the method described by Cagampang et al. [22]. Briefly, millet powder (100 mg) was put into 10 mL tube and immersed in 0.2 mL 95% ethanol containing 0.025% thymol blue. The tube was shaken in a boiling water bath at 100 rpm·min^−1^, with an added 2 mL 0.2 M KOH. After 8 min, the samples were placed on the desk at room temperature for 5 min. Then, they were cooled in an ice-water bath for 15 min. The tubes were laid horizontally over ruled paper and the length of the gel was measured. Each test was set up with three biological replicates.

### 2.6. Grain Hardness and Color Index Determination

Grain hardness was measured by a hardness testing device (GWJ-2, LVBO Instrument Company, Hangzhou, China). A colorimeter was used to determine the grain color (X-rite VS50, Danaher Corporation, Washington, DC, USA). For each variety, 5 g of grains from 3 individual panicles were, respectively, collected and measured.

### 2.7. Data Collection and Statistical Analysis

The climate data collected in this study include: the temperature of ground-surface, air temperature, diurnal temperature range of ground-surface, diurnal temperature range, daily rain fall, daily sunshine hours, and altitude (ALT) in three planting environments from July to September 2018. These data were collected by and ordered from the China Meteorological Administration. The accumulated temperature of ground-surface (ATG) and ≥15 °C accumulated air temperature (AAT) were summed by calculating the accumulation of the mean daily temperature. The mean diurnal temperature range of ground-surface (MDTRG) and the mean diurnal temperature range (MDTR) were calculated by averaging the difference between the highest and lowest temperatures. The total rainfall (TRF) was calculated by summing the daily accumulated rainfall from July to September 2018. The daily average sunshine hours (DASSH) were calculated by the average daily accumulation of sunshine hours (Table 1).

K-means clustering analysis was carried out for the TPC of 295 accessions in foxtail millet (excluded 25 accessions with null values in the three planting environments) by the R function package (FactoExtra, https://rpkgs.datanovia.com/factoextra/index.html, accessed on 17 February 2022). According to the clustering results, accessions with similar distributions of TPC in the three planting environments were classified into the same group, and the eigenvalues of each group were calculated by the R function package WGCNA [23]. The correlation between each group and climatic factors was further calculated according to the Pearson correlation coefficient.

All data were collected and analyzed in EXCEL Version 2019. The statistical analysis was carried out with SPSS 21.0 software. R software (ggplot2, R verson 3.4.1, Bell Laboratories, Union County, NJ, USA) [24] was used to generate graphs. Correlation analyses of the different eating-quality traits were determined based on Pearson correlation coefficient. Through integrative analysis of eating-quality traits, the best varieties were screened by subordinate function method [25].

## 3. Results and Discussion

### 3.1. Characteristics of Total Protein Content Variation in Foxtail Millet

The grains of 301, 314 and 316 accessions were harvested from E1 (Datong city, 113°12′51″ E, 40°22′58″ N), E2 (Jinzhong city, 112°35′21″ E, 37°25′16″ N) and E3, (Jincheng city, 113°7′8″ E, 35°32′6″ N) in Shanxi province, respectively, at their mature stage in 2018, to gain a comprehensive understanding of the TPC distribution in our germplasm collection of foxtail millet accessions. For E1, the average and median TPCs were 9.51 and 9.44%, respectively, ranging from 4.81 to 14.56%; For E2, the average and median TPCs were 9.94 and 9.78%, respectively, ranging from 5.68 to 16.71%; For the E3 environment, the average and median TPCs were 10.53 and 10.49%, respectively, ranging from 5.50 to 15.34% (Figure 1a). The coefficient of variation (CV) of the TPC at E1 (18.5%) was higher than that at E2 (15.89%) and E3 (15.86%) (Table 2). The normal distribution test showed that the TPCs of these accessions conformed to a normal distribution in the three environments according to significant testing (Figure 1b–d). A further analysis of the skewness coefficient (SC) and kurtosis coefficient (KC) was also performed. SC analysis showed that the TPCs of the accessions grown at E2 were not evenly distributed around the mean and positively skewed (SC = 0.557, *p* = 0.08); the TPCs of the accessions grown at E1 and E3 were uniformly distributed around the mean (SC = 0.193 and 0.003, *p* = 0.2). The KC analysis showed that the TPC of the accessions grown at E2 showed a leptokurtic distribution (KC = 1.169, *p* = 0.08) while the TPCs of the accessions grown at E3 exhibited a mesokurtic distribution (KC = 0.078). The TPCs of the accessions grown at E1 demonstrated a slight platykurtic distribution (KC = −0.233) (Table 2).

Next, the individual variety performance of the TPC trait was examined. The crop yield and grain protein quality were subjected to genotype × environment interactions and compensatory effects between traits [26]. Firstly, the top performer in each environment was identified. We found B198 (14.56%), B328 (16.71%) and B118 (15.34%) possessed the highest TPC in E1, E2 and E3, respectively. Next, the varieties less-affected by environmental factors were also selected. We found 13 varieties (B10, B28, B85, B95, B112, B115, B117, B119, B139, B182, B243, B331 and B362) could maintain a relatively high TPC in all three environments. There were also eight accessions (B30, B203, B345, B367, B404, B413, B414 and B428) with stable but low TPC in the three environments. We found B331, especially, with high and relatively stable TPC (E1, 12.53%; E2, 12.45%; E3, 12.26%) in the three environments. The results suggested these varieties with stable TPC in the three environments have a strong environmental suitability. Breeding crop varieties with a high and stable TPC is fundamental to guaranteeing good quality in large-scale production. Meanwhile, the accessions with low but stable TPC should not be neglected, because they provide unique genetic resources with potential environmental suitability which can be introgressed into those lines with other excellent quality traits to generate elite lines. For example, a survey of barley grain protein concentration also included those cultivars with low but stable grain protein concentrations under various environmental conditions for breeding purposes [27].

### 3.2. Correlation Analysis of TPC and Climate Factors

K-means clustering analysis was first performed to group accessions with similar response patterns to the environmental factors in order to understand the environmental factors affecting the TPC. After some accessions with missing data under one or more environments were filtered out, 295 accessions of foxtail millet were classified into seven groups by K-means clustering (Figure 2). Of the groups, cluster 6 had most of the accessions (64 out of 295 accessions), accounting for 21.60% of the total accessions. The pattern of TPC in cluster 6 was higher in E2 than that in E3 and E1 (E2 > E3 > E1). The second major group was cluster 5, with 61 accessions, accounting for 20.68% of the total accessions. Its TPC showed a gradually decreasing trend in the three planting environments (E3 > E2 > E1). The smallest group was cluster 7, with only 24 accessions, accounting for 8.14% of the total accessions, and its TPC showed relatively stable trends across the three planting environments.

To further investigate the effect of individual climate factors on the TPC, the correlation relationships between TPC and ATG, AAT, MDTRG, MDTR, TRF, DASSH, and ALT were analyzed (Figure 3). The results revealed that the TPC of 61 accessions in cluster 5 was positively correlated with ATG (*p* < 0.05) and MDTGR (*p* < 0.001) but significantly negatively correlated with MDTR (*p* < 0.01). The TPC of 88 accessions in clusters 6 and 7 was significantly positively correlated with AAT (*p* < 0.05, *p* < 0.01), and the TPC of 32 accessions in cluster 1 showed a significant positive correlation with DASSH (*p* < 0.01); however, the TPC of the remaining groups did not show any correlation with any of these climate factors.

These results suggested that the TPC trait of some accessions could be strongly subject to the effect of the climate factors such as AAT, ATG, MDTR, MDTRG and DASSH. At the same time, some accessions were less affected by the climate factors, implying that they have potential in environmental suitability. For environmental factors, it has been reported that a higher altitude planting environment can enhance the yield and grain protein content of wheat and foxtail millet [26,28]. However, our results showed that the different ALTs did not significantly affect the TPC of the different genotypes of foxtail millet, although 149 accessions showed higher TPCs at relatively lower ALT planting environments, with a nonsignificant correlation coefficient greater than −0.9. Geographical distribution is an important environmental factor affecting crop yield and quality [29]. Meanwhile, it has also been accepted that foxtail millet is suitable for wide geographical distribution without compromising the quality and yield. On the other hand, our field experiments were limited to the Shanxi Province only. More field experiments with larger ALT variations would provide useful insight into the geographical factor effect on foxtail millet quality in future. In soybean, the TPC of 763 samples from 2010 to 2013 was positively correlated with accumulated temperature and the mean daily temperature, but was negatively correlated with hours of sunshine and the diurnal temperature range [30]. This was consistent with our results, indicating temperature as an important environmental factor for most crops. Another report showed that temperature and rainfall were spatially associated with protein variability in soybean [31]. However, our results showed that the TPC of these accessions was not significantly correlated with the TRF. One possible reason is that the TRF in the three planting environments did not show a big variation. Another explanation is that foxtail millet is well known to be drought tolerant; therefore, TRF is a minor environment factor for the TPC trait in foxtail millet.

### 3.3. Amino Acid Composition and Evaluation in Foxtail Millet Grains

To further evaluate other quality traits (amino acid composition, starch content, gel consistency, grain hardness and color index) of the accessions with a stable TPC, 10 representative accessions with a CV < 15% (Appendix A) were selected. Among these accessions, five accessions (B28, B95, B139, B117 and B331) had high protein content (>12%) and another five accessions (B203, BB404, B428, B414 and B413) had low protein content as control (<8%).

Firstly, the amino acid compositions of those 10 accessions were measured and analyzed. Seventeen amino acids could be detected by our instrument in the grains of foxtail millet, including nine essential amino acids (histidine, His; isoleucine, Ile; leucine, Leu; lysine, Lys; methionine, Met; phenylalanine, Phe; threonine, Thr; tryptophan, Trp; valine, Val). These amino acid compositions were clustered into two groups by hierarchical clustering analysis according to their content in different accessions as shown in Figure 4a. Group I contained glutamic acid (Glu), leucine (Leu), alanine (Ala), asparagine (Asp) and proline (Pro), which were found to have relatively high contents in foxtail millet grains. The remaining 12 amino acids belonged to Group II. The content of Glu was the highest among all the amino acids detected in all accessions, ranging from 82.6 to 197.8 mg·g^−1^ of protein extract. Meanwhile, the content of cystine (Cys) was the lowest, ranging from 2.2~7.2 mg·g^−1^ of protein extract. The results showed that there is a significant difference in the content of amino acid components between high-TPC and low-PC accessions. Specifically, the contents of Leu and Glu in high-TPC accessions were 1.75- and 1.64-fold more than those of low-TPC accessions, which were larger than the differences in their TPC, suggesting that those amino acids contribute more to the TPC difference.

An *AAS* evaluation was performed to assess the essential amino acid score. The *AAS* scores for His, Ile, Leu, Phe + Tyr, Thr and Val in the grains of foxtail millet were significantly higher than their scores in the FAO/WHO/UNU standards (>100%). Of these amino acids, Leu had the highest *AAS* scores in the E1, E2 and E3 planting environments, with values ranging from 83.64 to 199.28%, from 78.78 to 180.25% and from 71.93 to 176.40%, respectively. Subsequently, the second highest *AAS* score was observed for Phe + Tyr in the E1, E2 and E3 planting environments, with values ranging from 94.39 to 171.43%, from 96.30 to 174.25% and from 91.73 to 176.89%, respectively. Based on our analysis, the limiting amino acid was Lys, with *AAS* scores of 23.97~40.31% in the three planting environments (Figure 4b; Appendix A).

The protein quality of 10 accessions from 3 environments were tested next. The *EAAI* is used to evaluate the protein quality of food by measuring the geometric mean of *EAAs* in proteins. The closer the *EAAI* value is to 100 indicates the closer the *EAA* composition is to the standard protein in the tested protein sample, suggesting a higher nutritional value in the tested sample. For E1, 4 out of 5 high-TPC accessions had high-*EAAI* scores (more than 90%). Among those four accessions, B139 had the highest EAAI score of 103.31%. For E2, all five high-TPC accessions had high *EAAI* scores (more than 90%). Specially, B28 had the highest *EAAI* score of 108.2%. For E3, four of five accessions had high *EAAI* scores (more than 100%). Among them, B331 had the highest *EAAI* score of 107.11% (Appendix A). In summary, accessions B139, B28 and B331 had good protein quality in all three environments according to the *EAAI* score. In contrast, the varieties with low TPC had low *EAAI* scores in the three environments, ranging from 57.30 to 71.83%, which is not surprising (Appendix A).

The greater *AAS* and *EAAI* values of the crop grains suggested that they have a balanced amino acid composition and therefore higher protein quality [32]. Previously, a report on the amino acid profiles of foxtail millet showed that Glu, Leu, Pro and Ala were the principal amino acids, and Lys was a limit amino acid [33]. Our results showed that the *AAS* values of seven *EAAs*, except Lys, in most foxtail millet accessions were more than 100%. This result suggested that foxtail millet grains could be a good-quality protein source for the human diet and animal feedstock, which is consistent with previous reports. However, our study also confirmed that Lys was still the first limiting amino acid in foxtail millet grains, similar to the scenario in other cereal crops.

The free amino acids accounting for the different tastes in the grains of 10 foxtail millet accessions were further evaluated (Table 3). The fresh, bitter and sweet amino acids accounted for 29.10 to 30.35%, 26.77 to 27.77% and 41.88 to 43.76% of the total amino acid species, respectively. Specifically, the accession containing the most of the three flavor amino acids was B139, with 30.35% fresh amino acids, 27.77% sweet amino acids, and 41.88% bitter amino acids. The differences in content of good (delicious and sweet amino acids) and bitter amino acids were analyzed for the 10 accessions of foxtail millet grains. The results showed that B139, with more good taste amino acids (16.24% more than bitter amino acids) could be a fresher accession because of its higher fresh and sweet amino acid contents. In contrast, B413 was less tasty according to our flavor-related amino acid analysis (only 12.48% difference in good and bitter taste amino acids).

The umami taste is elicited by L-Glu and L-Asp through the synergistic effect of 5′-nucleotide, guanosine-5′-monophosphate (GMP) and inosine-5′-monophosphate [34]. In our results, the contents of Glu and Asp were higher than other amino acids in the foxtail millet grains. Through a taste analysis of these amino acid compositions, we found that the ratio of fresh amino acids plus sweet amino acids (56.24~58.12%) was higher than that of bitter amino acids in all foxtail millet accessions (41.88~43.76%). In the meantime, the difference between the ratio of fresh amino acids plus sweet amino acids and bitter amino acids showed a slight variation (12.48~16.24%), suggesting good taste quality in foxtail millet. However, there was higher bitter amino acid content, accounting for 41.88~43.76% of the total amino acid content, which could be a bottleneck for breeding for excellent taste quality. It could be a key breeding target trait to reduce the bitter amino acid content and improve the taste quality of foxtail millet in the future.

### 3.4. Eating-Quality Evaluation and Correlation Analysis of Foxtail Millet

In order to screen the best variety with a stable TPC and other eating-quality traits, the total starch content (TSC), grain hardness index (GHI), gel consistency (GC) and grain color index (GCI) were analyzed in the 10 accessions of foxtail millet mentioned above. The TSC of these accessions ranged from 50.96 to 64.11, 51.71 to 60.51 and 47.45 to 61.09 g·100 g^−^^1^ in the E1, E2, and E3 environments, respectively (Appendix A). It was also noted that the mean value of TSC at E2 was higher than E1 and E3. Noticeably, the TSC of B331 was the highest with 55.63 g·100 g^−1^ TSC. B331 and B28 also showed high TSC in accessions with high TPC, suggesting they have good eating quality.

For GHI in the other varieties, B139 had a higher mean GHI, at 3.8 N in the three environments (Appendix A). The lowest mean GHI was in B95, at 1.73 N. There was a greater variation in GHI in B28 and B413 than in the other accessions, with 17.22 and 20.73% CV, respectively, in the three environments.

The GC of the 10 accessions in the E1, E2 and E3 environments ranged from 69.89 to 123.92, 68.14 to 117.62 and 63.72 to 111.48 mm, respectively (Appendix A). Of those accessions, B95 had the lowest mean GC in the three environments at 67.25 mm. However, the mean GC of B414 in the three environments was the highest at 116.73 mm. The CV of GC for B28 in the three environments was the highest at 10.17%, but that for B428 was the lowest at 0.6%.

For the GCI analysis, the b values of the 10 foxtail millet accessions in the E1, E2 and E3 environments ranged from 36.09 to 47.33, 35.60 to 48.23 and 36.64 to 46.50, respectively (Appendix A). The mean b value for B95 in the three environments was the lowest at 36.11. The CV of the b value for B117 in the three environments was the highest, at 5.53%, but that for B95 was the lowest, at 1.16%.

The L values of the 10 foxtail millet accessions in the E1, E2 and E3 environments ranged from 59.5 to 67.37, 58.39 to 66.42, and 58.61 to 65.60, respectively (Appendix A). B28 had the lowest mean L value in the three environments at 59.39; however, B413 had the highest mean L value at 65.95. The CV of the L value for B95 in the three environments was the highest, at 4.98%, and that for B404 was the lowest, at 0.51%.

The Pearson correlation analysis was performed to reveal the potential relations of these eating-quality traits. Based on our analysis, GC showed significant negative correlation with TPC but positively correlated with TSC (*p* < 0.001) in the E1, E2 and E3 planting environments (*p* < 0.001) (Figure 5). The L and b values of the GCI were positively correlated with GC (*p* < 0.05) but significantly negatively correlated with TPC in the E2 planting environment (*p* < 0.01). Moreover, GC was significantly negatively correlated with TPC in the E2 planting environment (*p* < 0.001). In the E3 environment, the L value of the GCI was positively correlated with TSC (*p* < 0.01) but negatively correlated with TPC (*p* < 0.05), while the b value of the GCI was positively correlated with GC (*p* < 0.01) and TSC (*p* < 0.05) but showed significantly negative correlation with TPC (*p* < 0.05). A similar correlation pattern of eating-quality traits was also observed in wheat and rice. In winter wheat, the investigation of the grain quality traits including grain protein content (GPC), grain starch content (GSC), and grain hardness (GH) in 372 accessions revealed that there was a significant negative correlation between the GPC and the other two traits—GSC and yield [35]. In rice, the amylose and protein contents were significantly negatively correlated with rice appearance, adhesiveness, and taste value, but positively correlated with hardness [36]. Taken together, these results suggest that the TPC significantly affects the eating-quality traits such as GC, GHI and TSC in cereal crops. Hence, it was important to further reveal the complex relationship between the TPC and eating-quality traits, which would be helpful in breeding superior quality varieties in foxtail millet with balanced nutritional and eating qualities.

### 3.5. Identification of Superior Eating and Protein Quality Variety

To screen the best variety with good protein and eating qualities, these accessions with high TPC were subjected to the integrative analysis of subordinative function value (U) of TSC, GHI, GC and GCI (b and L value) (Appendix A). The result showed that the highest U value of TPC and GC was the accession B331. However, the highest U value of GHI and GCI (L value) was the accession B95. In addition, B117 and B331 had higher U value of GCI (b value). By calculating the U mean of TPC, GHI, GC and GCI, the order of these varieties in the aspect of eating and protein qualities were B331, B28, B139, B117 and B95. For foxtail millet, the b value of grain color represents yellow color in grain appearance quality. The more yellow color in grain appearance were more popular with consumers in China because of its enriched carotenoids and the impression of more healthy compounds [7]. The superior eating quality was positively correlated with GC of grains, but negatively correlated with GHI in rice [37]. However, the high GC and GHI were desired quality in foxtail millet. Taken these results together, we suggest that the ac cession B331 is the best variety with high TPC and good eating quality, which could be served as core genetic resource for foxtail millet in future breeding programs.

## 4. Conclusions

In this study, we performed a survey of the protein content, amino acid composition and eating quality of a germplasm collection of foxtail millet accessions in different environments. Our results revealed several accessions with stable protein contents under different environments, suggesting their stability for wide cultivation. The correlation analysis further revealed that the protein content of foxtail millet accessions was positively correlated with climate factors such as ATG, MDTRG, AAT and DASSH, implying that the protein content trait of the foxtail millet grains was affected by environmental and genotypic interactions.

The amino acid composition analysis further provided some insight into nutritional value and flavor traits. We found that high protein content accessions also had a better essential amino acid index, providing more nutritional value for human beings and animals. Lys is still the limiting amino acid in foxtail millet grains, similar to the scenario in other cereal crops. Interestingly, Leu and Phe + Tyr contribute in higher proportions to the high-protein content varieties than the low-protein content varieties. Moreover, bitter amino acid contents were relatively high in all varieties, which could be a bottleneck for future eating-quality improvement.

The following eating-quality analysis of several traits suggest that the variance of GHI among the varieties was less than those of GC and SC. Additionally, we found that TPC is negatively correlated with GC and TSC, which is common in other cereal crops such as wheat and rice. The subordinative analysis suggested that B331 could be the best variety with a high protein content and superior eating quality.

Taken together, this study provides essential nutritional and eating-quality data on our germplasm collection of foxtail millet varieties, which will benefit the growing community of foxtail millet researchers and breeders, since foxtail millet is an emerging healthy, gluten-free cereal crop. Moreover, this preliminary research also explored and supplied a resource of core genetic materials for further introgression breeding.

## Figures and Tables

**Figure 1 foods-12-00031-f001:**
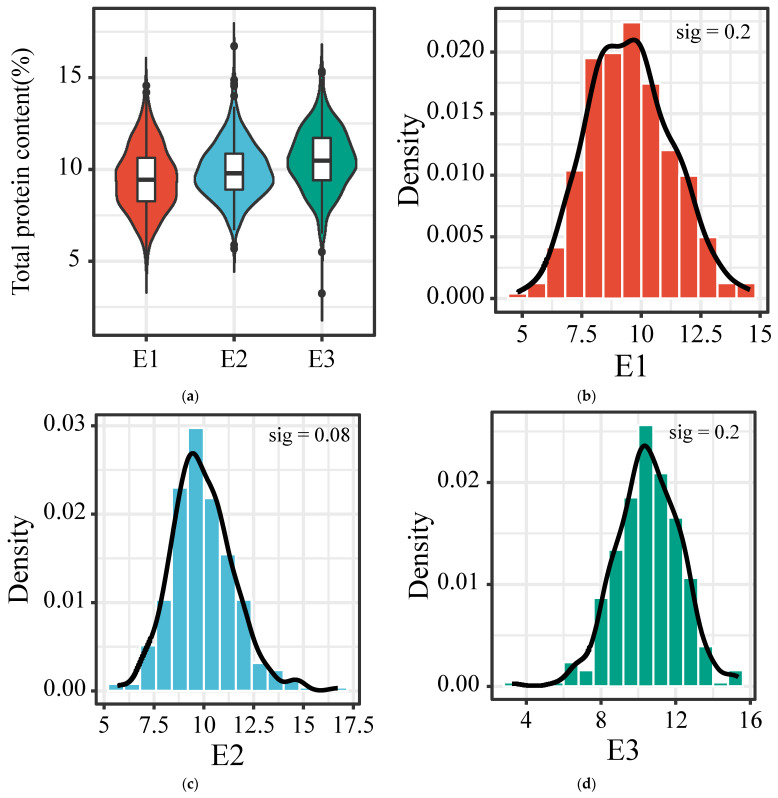
Distribution of TPC in a germplasm collection of foxtail millets grown at three experimental fields E1 (Datong city, 113°12′51″ E, 40°22′58″ N), E2 (Jinzhong city, 112°35′21″ E, 37°25′16″ N) and E3, (Jincheng city, 113°7′8″ E, 35°32′6″ N) in Shanxi province, China. (**a**) Grain TPC distribution characteristics of 301, 314 and 316 accessions harvested from E1, E2 and E3, respectively. (**b**–**d**) Histograms showing the normal distribution of TPC in three environments (E1, B; E2, C; E3, D) of foxtail millet. Significant test (sig) was also performed and labeled in each histogram.

**Figure 2 foods-12-00031-f002:**
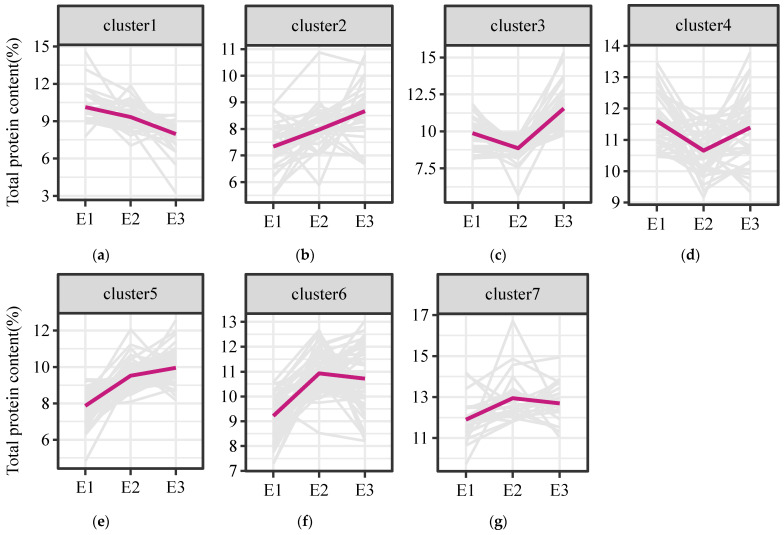
Distribution trend of TPC of foxtail millet in seven clusters (**a**–**g**) under three environments. K-means clustering analysis was performed in 295 accessions of foxtail millet under three environments. The accessions were grouped into 7 clusters with different trends of TPC in E1, E2, and E3.

**Figure 3 foods-12-00031-f003:**
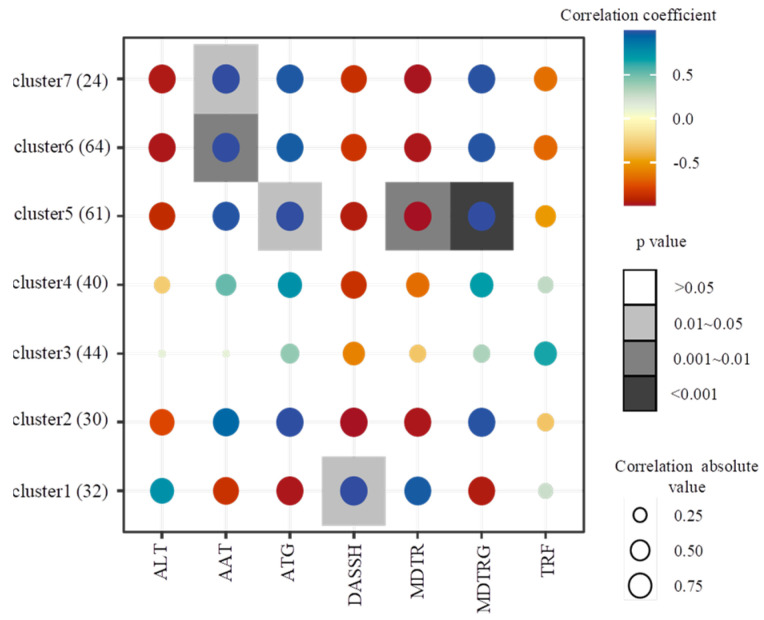
Correlation analysis between different clusters and climate data. ATG: accumulated temperature of ground-surface; AAT: accumulated temperature; MDTRG: mean diurnal temperature range of ground-surface; MDTR: mean diurnal temperature range; TRF: total rain fall; DASSH: daily average sunshine hours; ALT: altitude. The Pearson correlation coefficient was calculated by the ‘cor’ function in R software. The heatmap graph was plotted by ggplot2 in R software (R verson 3.4.1, Bell Laboratories, Union County, NJ, USA).

**Figure 4 foods-12-00031-f004:**
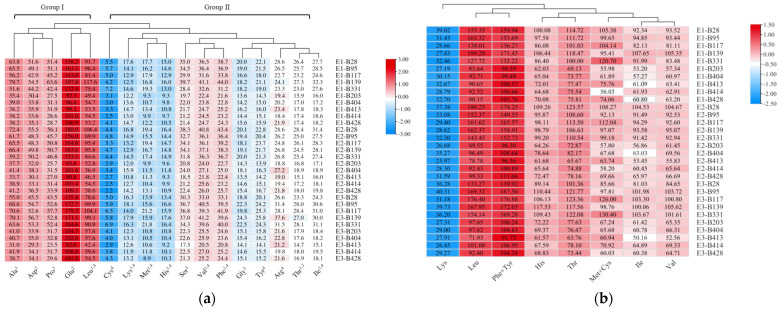
Clustering of amino acid content (mg·g^−1^) (**a**) and amino acid scores (*AAS*, %) of essential amino acids (**b**) in foxtail millet of 10 foxtail millet varieties in E1, E2 and E3. The 10 accessions include 5 accessions (B28, B95, B139, B117 and B331) with high protein content (>12%) and another 5 accessions (B203, BB404, B428, B414 and B413) with low protein content in foxtail millet grains. The amino acids, marked with “1” are essential amino acids (*EAA*), with “2” are fresh amino acids, with “3” are sweet amino acids, with “4” are bitter amino acids.

**Figure 5 foods-12-00031-f005:**
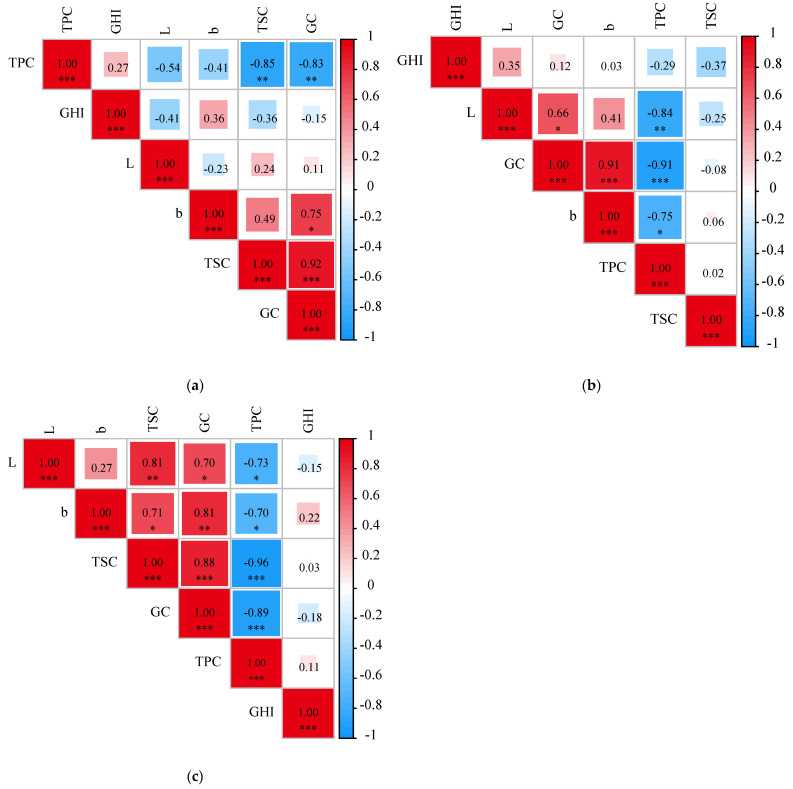
Correlation analysis between different environments and quality traits. (**a**): E1, (**b**): E2, (**c**): E3. (b: b value of grain color; L: L value of grain color; GC: gel consistency; TPC: total protein content, TSC: total starch content, GHI: grain hardness index). Note: * represents a *p* value < 0.05 by Duncan’s test; ** represents *p* < 0.01, *** represents *p* < 0.01.

**Table 1 foods-12-00031-t001:** The climate data in three plant environments.

Environment	ATG	AAT	MDTRG	MDTR	TRF	DASSH	ALT
E1	2186.30	1871.50	26.41	11.43	306.90	7.49	10,343.00
E2	2454.70	2031.70	27.99	10.81	279.80	7.17	7858.00
E3	2559.90	2048.60	28.55	10.60	299.40	6.80	8373.00

ATG: accumulated temperature of ground-surface; AAT: ≥15 °C accumulated air temperature; MDTRG: the mean diurnal temperature range of ground surface; MDTR: the mean diurnal temperature range; TRF: total rainfall; DASSH: daily average sunshine hours; ALT: altitude.

**Table 2 foods-12-00031-t002:** The statistical analysis of total protein contents among different accessions of foxtail millet in three environments.

Environment	Min (%)	Max (%)	Med	AV	SD	CV (%)	SC	KC	Sig (*p* > 0.05)
E1 (*n* = 301)	4.81	14.56	9.44	9.51 c	1.76	18.50	0.193	−0.233	0.2
E2 (*n* = 314)	5.68	16.71	9.78	9.94 b	1.58	15.89	0.557	1.169	0.08
E3 (*n* = 316)	5.50	15.34	10.49	10.53 a	1.67	15.86	0.003	0.078	0.2

Note: max: the maximum total protein content; min: the minimum total protein content; med: median; SD: standard deviation; AV: average total protein content; CV: coefficient of variable; SC: skewness coefficient; KC: kurtosis coefficient. Sig: significance test. Letter a–c represents the significance at *p* < 0.05 by Duncan’s multiple range test.

**Table 3 foods-12-00031-t003:** Flavor-related amino acid content in ten varieties of foxtail millet (%).

Variety	Fresh Amino Acids (%)	Sweet Amino Acids (%)	Bitter Amino Acids (%)	Difference Value between Fresh Amino Acids + Sweet Amino Acids and Bitter Amino Acids (%)
B28	29.79	27.67	42.53	14.93
B95	30.12	27.53	42.35	15.3
B117	30.06	27.60	42.34	15.32
B139	30.35	27.77	41.88	16.24
B331	29.72	27.21	43.07	13.86
B203	29.68	27.25	43.08	13.85
B404	29.10	27.57	43.34	13.33
B413	29.47	26.77	43.76	12.48
B414	30.11	27.16	42.74	14.53
B428	29.87	26.79	43.33	13.33

Fresh amino acids: Asp, Glu; sweet amino acids: Pro, Gly, Ser, Thr, Ala; bitter amino acids: Val, Leu, Ile, Met, Cys, Arg, His, Phe, Lys, Tyr.

## Data Availability

Data is contained within the article or Appendix A.

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
