# Peer review of "Total Protein Content, Amino Acid Composition and Eating-Quality Evaluation of Foxtail Millet (Setaria italica (L.) P. Beauv)"

_foods, 2022, doi:10.3390/foods12010031_

Round 1
Reviewer 1 Report
I commend the authors Siyu Hou, Yihan Men, Min Wei, Yijuan Zhang, Hongying Li, Zhaoxia Sun, Yuanhuai Han of the manuscript titled “Total protein content, amino acid composition and eating quality evaluation of foxtail millet (Setaria italica (L.) P. Beauv)” for their work on the evaluation of the total protein content (TPC), amino acid compositions and eating quality among 320 foxtail millet accessions in three environments. They aimed to identify the best variety with good eating quality
.
Before this paper is published, there are some things need to be addressed or corrected:
1. The English language need to be improved especially in the abstract. Its much better to be edited by native English speaker.
2. In the abstract, the abstract is poorly written and need be reconstructed and rewritten.
3. The introduction, in line 89, its better to start a new paragraph, so you need to split that big paragraph into 2 pieces
- New information need to be added about the current production indexes in China as well as world production and importance of this crop.
4. In the materials and methods :
- In line 113, Total protein content (TPC) determination, a citation for the method should be used .
- Line 124, citation to the method should be used.
- Line 160, further detailed description for the method should be added
5. In the results part:
- Figure 2 is not visible and need to be enlarged and sharpened
-
6. In the discussion part,
- Additional references and citation should be added to the parts of amino acid composition and analyses.
The conclusion part is necessary here and the prospect of the work as well
Overall the I give you major revision.
Author Response
- The English language need to be improved especially in the abstract. Its much better to be edited by native English speaker.
Response: Thank your helpful advice, we carefully revised these sentence in abstract. According to your proposal, we invited a native English speaker for revised our manuscript.
- In the abstract, the abstract is poorly written and need be reconstructed and rewritten.
Response: Thank your good advice, we carefully revised these sentence in abstract.Please see detail in L36-41 in red.
- The introduction, in line 89, its better to start a new paragraph, so you need to split that big paragraph into 2 pieces
- New information need to be added about the current production indexes in China as well as world production and importance of this crop.
Response: Thank your good advice. We carefully revised these paragraph and figure location. Please see page 3 L55-56,L60-64.
- In the materials and methods :
- In line 113, Total protein content (TPC) determination, a citation for the method should be used .
- Line 124, citation to the method should be used.
- Line 160, further detailed description for the method should be added
Response: Thank your good advices. We carefully revised these sentence and added the citation of these parts. Please see reference 19,20.
- In the results part:
- Figure 2 is not visible and need to be enlarged and sharpened
-Response: I’m sorry for the mistake. We carefully revised figure 2 and described details for it. Please see detail in page 11.
- In the discussion part,
- Additional references and citation should be added to the parts of amino acid composition and analyses.
The conclusion part is necessary here and the prospect of the work as well
Overall the I give you major revision.
Response: We carefully revised these sentences and rewritten this part. According to your advice, we combined the results and discussion and rewritten this part for conveniently reading. Please see detail in results and disscussions part.
Reviewer 2 Report
The paper aims to evaluate the quality of a large number of foxtail millet varieties grown in different localities under different environmental conditions. The data in the paper can be useful for creating a database, but on the other hand, there are serious issues that the authors must address and the paper requires extensive revision.
In the explanation of the motivation and the aim of the study, the authors stated that “we hope that these results show the differences in nutritional value and eating quality among different genotypes …” (Lines 98-99). It is quite expected that different varieties (genotypes), grown in different localities and under different climatic conditions, will have different characteristics. I don't see that it was necessary to prove this, except for the purpose of choosing the best varieties with good eating qualities. However, I do not see in the paper that the authors, based on their results, made a proposal as to which are the best varieties, given that the shortlist for a more detailed laboratory analysis was stable protein content, where five of the selected ten varieties (out of a total of 160 varieties if I counted correctly) were with a relatively low protein content.
Here we come to the key point that the authors must explain. What is the reason why stable protein content was chosen as a criterion for selecting 10 varieties for further analysis, again I repeat in the light of the fact that 5 of these varieties had a relatively low protein content and certainly do not belong to best varieties category at least in terms of nutritional profile. What is the reason that some varieties have stable protein content and others do not, i.e., what are the reasons why exactly these 10 show stability? Do the authors think that these 10 varieties with stable protein content should have an advantage over other varieties? Are the varieties selected for further analysis the ones with the best eating quality? I don't see that the authors identified which are the best varieties with good eating quality, they just showed that there are differences between the varieties, which was certainly expected (especially since they are grown in different locations and climatic conditions). Furthermore, it was not stated whether the agrotechnical measures were the same in all environments, especially the type and level of fertilization, because differences in protein content may be a consequence of that as well.
Abstract:
The paper provides data on individual characteristics of foxtail millet varieties, but in the end it details the characteristics of only 10 varieties grown under three environmental conditions. Therefore, I think it is too ambitious to claim that the paper provides an evaluating standard as stated by the authors in the abstract. First, I don't see that the authors proposed what should be the evaluating standard, and in order for something to get the status of a standard, it must go through a rigorous domestic or international procedure. Furthermore, the paper provides a certain database, but I do not see how it provides technical support to identify high-quality foxtail millet varieties suitable for planting, given that the criterion for selecting 10 varieties for a more detailed analysis was not high protein content, but protein stability, whereby five varieties with a relatively low protein content also showed this stability. The abstract should be rewritten taking into account these suggestions.
Materials and Methods
I am convinced that there is no need to describe standard methods such as Total protein content - micro Kjeldahl method and Starch content analysis in detail. It is enough to refer to the standard method (specify the number of the method) and source (AOAC, AACC, ICC, etc.).
Results
Instruction for authors allows the results and discussion to be given as two separate sections. However, if there is a separate section of the Results it should be concise. I suggest the authors to combine Results and Discussion, currently given as separate sections, into one section. Most of the section Results in this paper are a simple statement of numerical values ​​already shown in the tables (or figures). As such, they are unnecessarily repetitive, and without a discussion they have little significance. The paper would be much better structured if these two sections were combined into one.
Discussion
As suggested, the Discussion section needs to be rewritten by incorporating parts of the current Results section. The discussion, that is, the claims and conclusions presented in the discussion should be based on and backed up by referring to the obtained results of experimental tests.
Lines 383-386: The sentence "Hence, we hope to improve the content of delicious amino acids and sweet amino acids or reduce the content of bitter amino acids by genetically modifying foxtail millet to breed new varieties with good taste and new flavors in the future ". - I don't understand this sentence which is supposed to be a sort of conclusion of a manuscript. What do the authors mean by "we hope to improve the content of ..."? It is not clear to me in what way the results of this research will enable this, especially since in the following of the sentence authors talk about genetic modification that should result in new varieties with improved characteristics. I do not understand how these results will contribute to the genetic modification of foxtail millet. This part should be reformulated and explained more precisely.
Author Response
Reviewer #2
Comments and Suggestions for Authors
The paper aims to evaluate the quality of a large number of foxtail millet varieties grown in different localities under different environmental conditions. The data in the paper can be useful for creating a database, but on the other hand, there are serious issues that the authors must address and the paper requires extensive revision.
In the explanation of the motivation and the aim of the study, the authors stated that “we hope that these results show the differences in nutritional value and eating quality among different genotypes …” (Lines 98-99). It is quite expected that different varieties (genotypes), grown in different localities and under different climatic conditions, will have different characteristics. I don't see that it was necessary to prove this, except for the purpose of choosing the best varieties with good eating qualities. However, I do not see in the paper that the authors, based on their results, made a proposal as to which are the best varieties, given that the shortlist for a more detailed laboratory analysis was stable protein content, where five of the selected ten varieties (out of a total of 160 varieties if I counted correctly) were with a relatively low protein content.
Here we come to the key point that the authors must explain. What is the reason why stable protein content was chosen as a criterion for selecting 10 varieties for further analysis, again I repeat in the light of the fact that 5 of these varieties had a relatively low protein content and certainly do not belong to best varieties category at least in terms of nutritional profile. What is the reason that some varieties have stable protein content and others do not, i.e., what are the reasons why exactly these 10 show stability? Do the authors think that these 10 varieties with stable protein content should have an advantage over other varieties? Are the varieties selected for further analysis the ones with the best eating quality? I don't see that the authors identified which are the best varieties with good eating quality, they just showed that there are differences between the varieties, which was certainly expected (especially since they are grown in different locations and climatic conditions). Furthermore, it was not stated whether the agrotechnical measures were the same in all environments, especially the type and level of fertilization, because differences in protein content may be a consequence of that as well.
Response: Thank your helpful advice, we carefully revised these sentence and exactly focused on our aim in our study in further. According to your proposal, we added the reasons for selecting the best varieties with good eating qualities to the context. We also rearrange the results and disscussion part for clarified the analysis. First, we choose higher TPC accessions. Next, higher AAS and EAAI accessions were selecting from above 10 accession. Finally, the best taste quality accessions were choosed by subordinate function analysis. Please see detail in page 8-9, L214-250;page12 L285-306;page 13 L315-323; page 14-15 L347-386 and page 21 L 449-463.
Abstract:
The paper provides data on individual characteristics of foxtail millet varieties, but in the end it details the characteristics of only 10 varieties grown under three environmental conditions. Therefore, I think it is too ambitious to claim that the paper provides an evaluating standard as stated by the authors in the abstract. First, I don't see that the authors proposed what should be the evaluating standard, and in order for something to get the status of a standard, it must go through a rigorous domestic or international procedure. Furthermore, the paper provides a certain database, but I do not see how it provides technical support to identify high-quality foxtail millet varieties suitable for planting, given that the criterion for selecting 10 varieties for a more detailed analysis was not high protein content, but protein stability, whereby five varieties with a relatively low protein content also showed this stability. The abstract should be rewritten taking into account these suggestions.
Response: Thank your good advice, we carefully revised these sentence and exactly focused our aim in our study in further. According to your proposal, we rewritten this part and organized the language to describe the details.
Materials and Methods
I am convinced that there is no need to describe standard methods such as Total protein content - micro Kjeldahl method and Starch content analysis in detail. It is enough to refer to the standard method (specify the number of the method) and source (AOAC, AACC, ICC, etc.).
Response: Thank your good advises, we carefully revised these sentence and deleted these redundancy sentence in further.Please detail in page5 L 132-136.
Results
Instruction for authors allows the results and discussion to be given as two separate sections. However, if there is a separate section of the Results it should be concise. I suggest the authors to combine Results and Discussion, currently given as separate sections, into one section. Most of the section Results in this paper are a simple statement of numerical values ​​already shown in the tables (or figures). As such, they are unnecessarily repetitive, and without a discussion they have little significance. The paper would be much better structured if these two sections were combined into one.
Response: Thank your good advice, yes, we carefully revised these parts and combined the results and discussions in further. According to your proposal, we rewritten this part and organized the language to describe the details.
Discussion
As suggested, the Discussion section needs to be rewritten by incorporating parts of the current Results section. The discussion, that is, the claims and conclusions presented in the discussion should be based on and backed up by referring to the obtained results of experimental tests.
Lines 383-386: The sentence "Hence, we hope to improve the content of delicious amino acids and sweet amino acids or reduce the content of bitter amino acids by genetically modifying foxtail millet to breed new varieties with good taste and new flavors in the future ". - I don't understand this sentence which is supposed to be a sort of conclusion of a manuscript. What do the authors mean by "we hope to improve the content of ..."? It is not clear to me in what way the results of this research will enable this, especially since in the following of the sentence authors talk about genetic modification that should result in new varieties with improved characteristics. I do not understand how these results will contribute to the genetic modification of foxtail millet. This part should be reformulated and explained more precisely.
Response: Thank your good advises, yes, we carefully revised these sentence and combined the results and discussions in further. According to your proposal, we rewritten this part and additionally describe the details. Please see details in results and disscussion part. We extended the conclusion part either.Please see detail in page 22 L 479-488.
And other minor revises were marked in red.
Round 2
Reviewer 1 Report
Accepted for me
Author Response
I'm very appreciate for the reviewer's comments to improve the MS.
Reviewer 2 Report
This is a much improved version of the paper than the original. My concerns have been satisfied by the authors in their revision.
Author Response

(The authors gave the same response as above.)
